# Serum Levels of miR-122-5p and miR-125a-5p Predict Hepatotoxicity Occurrence in Patients Undergoing Autologous Hematopoietic Stem Cell Transplantation

**DOI:** 10.3390/ijms25084355

**Published:** 2024-04-15

**Authors:** Damian Mikulski, Kacper Kościelny, Izabela Dróżdż, Grzegorz Mirocha, Mateusz Nowicki, Małgorzata Misiewicz, Ewelina Perdas, Piotr Strzałka, Agnieszka Wierzbowska, Wojciech Fendler

**Affiliations:** 1Department of Biostatistics and Translational Medicine, Medical University of Lodz, 92-215 Lodz, Poland; damian.mikulski@umed.lodz.pl (D.M.); kacper.koscielny@stud.umed.lodz.pl (K.K.); grzegorz.mirocha@stud.umed.lodz.pl (G.M.); ewelina.perdas@umed.lodz.pl (E.P.); 2Department of Hematooncology, Copernicus Memorial Hospital in Lodz, 93-513 Lodz, Poland; 3Department of Clinical Genetics, Medical University of Lodz, 92-215 Lodz, Poland; izabela.drozdz@umed.lodz.pl; 4Department of Hematology, Medical University of Lodz, 92-215 Lodz, Poland; mateusz.nowicki@umed.lodz.pl (M.N.); malgorzata.misiewicz@umed.lodz.pl (M.M.); piotr.strzalka@stud.umed.lodz.pl (P.S.); agnieszka.wierzbowska@umed.lodz.pl (A.W.); 5Department of Hematology and Transplantology, Copernicus Memorial Hospital in Lodz, 93-513 Lodz, Poland

**Keywords:** ASCT, hematopoietic stem cell transplant, hepatotoxicity, miR-122-5p, miR-125a-5p, liver injury

## Abstract

Hepatic complications are an acknowledged cause of mortality and morbidity among patients undergoing hematopoietic stem cell transplantation. In this study, we aimed to evaluate the potential role in the prediction of liver injury of five selected microRNAs (miRNAs)—miR-122-5p, miR-122-3p, miR-15b-5p, miR-99b-5p, and miR-125a-5p—in the setting of autologous hematopoietic stem cell transplantation (ASCT). A total of 66 patients were included in the study: 50 patients (75.8%) with multiple myeloma (MM) and 16 (24.2%) with lymphoma. Blood samples were collected after the administration of the conditioning regimen, on the day of transplant (day 0). The expression levels of selected miRNAs were quantified by reverse transcription-quantitative polymerase chain reaction (RT-qPCR) using the miRCURY LNA miRNA Custom PCR Panels (QIAGEN). In a multivariate logistic regression analysis adjusted for age, sex, and the administered conditioning regimen, two miRNAs, hsa-miR-122-5p (odds ratio, OR 2.10, 95% confidence interval, CI: 1.29–3.42, *p* = 0.0029) and hsa-miR-125a-5p (OR 0.27, 95% CI: 0.11–0.71, *p* = 0.0079), were independent for hepatic toxicity occurrence during the 14 days after transplant. Our model in 10-fold cross-validation preserved its diagnostic potential with a receiver operating characteristics area under the curve (ROC AUC) of 0.75, 95% CI: 0.63–0.88 and at optimal cut-off reached 72.0% sensitivity and 74.4% specificity. An elevated serum level of miR-122-5p and decreased level of miR-125a-5p on day 0 are independent risk factors for hepatotoxicity in ASCT recipients, showing promise in accurately predicting post-ASCT complications. Identifying patients susceptible to complications has the potential to reduce procedure costs and optimize the selection of inpatient or outpatient procedures.

## 1. Introduction

High-dose chemotherapy with autologous hematopoietic stem cell transplantation (ASCT) is the most common type of hematopoietic stem transplantation, exceeding 50,000 procedures performed worldwide annually [1]. The most common indication for this remains multiple myeloma (MM) and other plasma cell disorders, followed by non-Hodgkin lymphomas (NHLs) and Hodgkin lymphoma (HL). Recently, investigations into the safety and efficacy of outpatient and early discharge strategies for ASCT have emerged [2,3,4]. Despite the increasing popularity of these approaches, there is a notable absence of clinically validated biomarkers for accurately predicting post-ASCT complications or identifying high-risk individuals. Predicting which patients are prone to complications therefore holds potential for reducing costs and optimizing the selection of inpatient or outpatient procedures.

Liver complications constitute one of the leading causes of both mortality and morbidity in individuals undergoing hematopoietic stem cell transplantation. Major contributors to hepatic complications in transplant recipients include cytostatic drugs in conditioning regimens, acute and chronic graft-versus-host disease, sinusoidal obstruction syndrome, and various infections, among other factors [5,6,7]. In patients undergoing allogenic transplantation, acute and chronic graft-versus-host disease (GvHD) are the most common causes of liver injury, often leading to inferior survival of affected patients [8]. In ASCT recipients, the leading cause of hepatotoxicity, which, depending on the type of therapy, may affect up to 40% of recipients, is the conditioning regimen [5].

MicroRNAs (miRNA) are a group of short, non-coding ribonucleic acid (RNA) molecules, typically ranging from 22 to 24 nucleotides, presenting themselves as promising biomarkers [9]. The biological stability of miRNAs in the extracellular space has been validated, underscoring their durability and ability to resist rapid degradation, establishing them as robust and easily clinically applicable biomarkers [10]. miRNAs were extensively studied in the context of hepatotoxicity and drug-induced liver injury (DILI) [11,12,13]. However, the role of miRNAs in the setting of conditioning regimen hepatotoxicity in ASCT recipients is unknown.

In this study, we aimed to evaluate the potential role of five selected miRNAs: miR-122-5p [11,12,13], miR-122-3p [11,12,13], miR-15b-5p [14,15,16], miR-99b-5p [17], and miR-125a-5p [17], all previously reported to be deregulated in the drug-induced liver injury and/or other liver disorders, in the specific setting of ASCT.

## 2. Results

### 2.1. Study Group Characteristics

A total of 66 patients were included in the analysis, comprising 34 women (51.5%) and 32 men (48.5%). The median age at ASCT was 59 years (interquartile range, IQR: 49–65). MM was diagnosed in 50 patients (75.8%), while the remaining 16 (24.2%) had lymphoma, including eight cases (12.1%) of HL, four cases (6.1%) of mantle cell lymphoma (MCL), two cases (3%) of diffuse large B-cell lymphoma (DLBCL), and one case (1.5%) each of angioimmunoblastic T-cell lymphoma (AICL) and anaplastic large cell lymphoma (ALCL). In MM patients, Immunoglobulin G (IgG) kappa was the most prevalent paraprotein (24; 48%), followed by IgG lambda (11; 22%). According to the International Staging System (ISS), 15 patients (30%) had stage III. Most patients with lymphoma had high Ann Arbor stage III (4 patients, 33.3%) and IV (5 patients, 41.7%). Most MM patients received one line of induction treatment before ASCT (31; 62%). In patients with lymphoma, most patients (7; 43.8%) received four lines prior to ASCT, whereas two patients with MCL received ASCT as consolidation after the first line of therapy. Most patients with MM received a Mel-200 conditioning regimen (32; 48.5%), whereas the rest received a reduced dose of melphalan (18; 27.3%). Among patients with lymphoma, the BeEAM (Bendamustine, Etoposide, Cytarabine, Melphalan) regimen was most commonly used (11; 17.2%), followed by BEAM (Carmustine, Etoposide, Cytarabine, Melphalan) (5; 7.8%). Concurrent hepatitis B virus (HBV) or hepatitis C virus (HCV) infection was present in eight patients (12.1%).

At admission, two (3.0%) patients had elevated alanine aminotransferase (ALT) and aspartate aminotransferase (AST) levels above the upper limit. After the conditioning chemotherapy (day 0), four patients (6.1%) were identified as having grade 1 hepatic toxicity. Subsequently, 14 patients (21.2%) developed liver injury during the first week after ASCT, followed by 16 patients (24.2%) in the second week after ASCT and 15 patients (22.7%) at the time of discharge. The median time to hepatic toxicity occurrence since ASCT was 6.5 days (IQR: 2.5–11 days). Overall, the majority of patients (34; 51.5%) had hepatic toxicity during hospitalization, and three patients (8.8%) had grade 2 hepatic toxicity according to the Seattle Criteria. The detailed characteristics of the study cohort are provided in Table 1.

### 2.2. miRNAs Expression Levels in ASCT Patients in Relation to Liver Injury

Overall, exploratory analysis using heatmap and hierarchical clustering yielded two clusters of patients with distinct miRNA profiles and frequencies of liver injury occurrence (Figure 1 and Appendix A).

Receiver operating characteristics (ROC) curve analysis showed that only hsa-miR-122-5p was considered discriminative for liver injury with area under the curve (AUC) 0.67, 95% CI: 0.53–0.82, *p* = 0.0186 (Appendix A). Overall, in patients with higher hsa-miR-122-5p serum levels, higher fold changes in ALT and AST levels compared to baseline (at admission) were observed across the ASCT procedure (Figure 2).

### 2.3. Factors Associated with Liver Injury

In the univariate analysis, none of the clinical variables significantly impacted the occurrence of hepatic toxicity (Table 2). There was a higher inclination toward hepatotoxicity in patients receiving the B(e)EAM conditioning regimen (odds ratio, OR: 1.58, 95% confidence interval, CI: 0.47–5.35) compared to those receiving Mel-200; however, this difference did not reach statistical significance (*p* = 0.4595). Importantly, none of the traditional hepatic markers measured upon admission, including albumin, AST, ALT, bilirubin, ALP, and GGTP, demonstrated an association with ASCT-related hepatotoxicity (Table 2). Among the analyzed miRNAs, only hsa-miR-122-5p emerged as a significant predictor, indicating a higher risk of hepatotoxicity with an OR of 1.59 (95% CI: 1.09–2.32, *p* = 0.0164).

In the next step, a multivariate logistic regression model was constructed utilizing two miRNAs with *p* < 0.1 in univariate analyses—hsa-miR-122-5p and hsa-miR-125a-5p adjusted for age, sex, and administered conditioning regimen. In this model, both miRNAs were significant predictors of liver injury occurrence—hsa-miR-122-5p (OR 2.10, 95% CI: 1.29–3.42, *p* = 0.0029) and hsa-miR-125a-5p (OR: 0.27, 95% CI: 0.11–0.71, *p* = 0.0079) (Table 3). Importantly, in analysis considering only MM patients, similar results were obtained (Appendix A).

Our model in 10-fold cross-validation preserved its diagnostic potential with an ROC AUC: 0.75, 95% CI: 0.63–0.88 and at optimal cut-off reached 72.0% sensitivity and 74.4% specificity (Figure 3).

### 2.4. Prediction of miRNAs Target Genes and KEGG Pathway Analysis

Overall, two miRNAs—miR-122-5p and miR-125a-5p—included in the final multivariate model targeted a total of 870 genes—miR-122-5p—610 genes, miR-125a-5p—270 genes (Appendix A and Appendix A). Ten genes were targeted by both investigated miRNAs. In the next step, functional enrichment analysis of identified targets was performed using hypergeometric testing with the Kyoto Encyclopedia of Genes and Genomes (KEGG) database (Appendix A). Overall, KEGG pathway analysis demonstrated that miRNAs deregulated in patients with liver toxicity among AHSCT recipients were enriched for genes associated with liver function and diseases, including ErbB (erythroblastic oncogene B) signaling pathway, Neurotrophin signaling pathway, Insulin signaling pathway, mTOR (mammalian target of rapamycin) signaling pathway, Hepatitis C, TGF-beta (transforming growth factor beta) signaling pathway, Jak-STAT (Janus kinase/signal transducers and activators of transcription signaling pathway), NOD-like (nucleotide-binding oligomerization domain-like receptors) receptor signaling pathway, MAPK (mitogen-activated protein kinase) signaling pathway, and VEGF (Vascular endothelial growth factor) signaling pathway.

## 3. Discussion

This study assessed the potential role of serum free circulating miRNAs in predicting liver injury in ASCT recipients. Notably, serum expression levels of miR-122-5p and hsa-miR-125a-5p at the day of transplant were identified as independent prognostic factors of hepatotoxicity occurrence in multivariate analysis. Significantly, toxicity manifested at a median of 6.5 days after evaluating the expression of miRNAs, which suggests that this finding could be utilized in clinical settings to take preventive measures. In our study, we focused on the liver toxicity occurrence in the early post-transplant period (since day 0 up to 14 days after transplant), as the majority of drug toxicity and sinusoidal obstruction syndrome (SOS) events occur in this period [18,19].

MiR-122 is an example of tissue-specific miRNA, as it is highly expressed in the liver tissue. It dominates the hepatic miRnome, accounting for 70% of the total miRNA in the liver [20,21,22]. A high serum miR-122-5p expression level was previously reported as a hallmark of acetaminophen-induced liver injury and HBV infection [23,24]. MiR-122 is also involved in HCV replication and interacts with the binding sites of HCV RNA, leading to a miR-122–HCV complex that protects the HCV genome from nucleolytic degradation or host innate immune responses [22,25,26]. These observations lead to the development of miravirsen antisense oligonucleotide complementary to the 5′ region of miR-122, which could inhibit the function of miR-122. The drug was evaluated in a phase 2 trial in patients with chronic HCV genotype 1 infection and yielded promising results. It was also reported that serum levels of miR-122 and miR-200a are higher in HIV-1-infected individuals treated with antiretroviral therapy before the development of fatal liver disease and may act as potential biomarkers in this setting [27]. Recently, elevated serum miR-122 expression has also been reported in patients with antituberculosis drug-induced liver injury compared to individuals treated with antituberculosis therapy without hepatotoxicity [28]. Taken together, miR-122 is an established biomarker for liver injury caused by various factors. Compared to other miRNAs, the high abundance of miR-122-5p within hepatocytes, its high degree of tissue selectivity, and its release from damaged cells may account for the increase in serum mir-122 expression level. We add to this existing knowledge and validate miR-122 as a hepatotoxicity biomarker in ASCT recipients. Furthermore, in this clinical setting, the effect of miR-122 was independent of the type of conditioning therapy (Mel-200 or B(e)EAM), serving as a potentially universal liver injury predictor.

In our study, a higher expression level of miR-125a-5p was acting protectively against the occurrence of liver injury. Overall, miR-125a-5p was reported to be deregulated in liver diseases. Specifically, serum expression of miR-125a-5p was increased in liver fibrosis, whereas in hepatocellular carcinoma, it was decreased compared to the healthy controls [29]. In addition, the miR-125a-5p level is increased in liver tissue in patients with HBV chronic infection and higher levels were associated with higher disease burden [30]. Recently, in a rat model, it was shown that miR-125a inhibited hepatocyte proliferation and liver regeneration by regulating the STAT3/p-STAT3/JUN/BCL2 axis [31]. Taken together, we may hypothesize that patients who will not develop liver injury during ASCT may have higher expression levels of miR-125a-5p, potentially facilitating a decreased need for liver regeneration after the conditioning regimen, whereas individuals at risk of hepatotoxicity may have reduced serum levels, enabling further hepatocyte proliferation. However, these observations should be confirmed in studies conducted on larger populations.

To investigate the potential mechanisms of miR-122-5p and miR-125a-5p in hepatotoxicity following ASCT, we conducted target gene prediction analysis. However, it is important to interpret bioinformatic predictions of miRNA function based on serum levels cautiously. This is because observed patterns may reflect an organism-level response to chemotherapy, and their expression could be specific to certain tissues. Nevertheless, our findings reveal that miRNAs deregulated in ASCT recipients experiencing liver toxicity were enriched for genes previously linked to liver function and diseases. For example, the ErbB signaling pathway is regarded as a crucial defense system for the liver when faced with acute injury. However, growing evidence indicates that its prolonged activation may contribute to the liver transition to a neoplastic state [32]. The epidermal growth factor receptor (EGFR), alternatively referred to as ErbB1, is a transmembrane protein receptor possessing tyrosine kinase activity. Hepatocytes exhibit notably elevated levels of ErbB1, and the expression of the majority of ErbB ligands rises during liver tissue damage and regeneration, as evidenced in rodent models of partial hepatectomy and experimental liver injury [33]. We have also found that the Neurotrophin signaling pathway was significantly enriched in our study. It was shown that during fibrotic liver injury, nerve growth factor (NGF) expression increases and may control the quantity of activated hepatic stellate cells by triggering apoptosis [34]. Moreover, NGF and its associated receptor are significant in influencing the pathophysiology of the intrahepatic biliary epithelium during liver tissue remodeling in cirrhosis and the progression of hepatocellular carcinoma [35]. In addition, increasing evidence suggests that neurotrophic factors can influence every phase of non-alcoholic fatty liver disease. Changes in both the autonomic nervous system and neurotrophic factors have been observed in patients and murine models of non-alcoholic fatty liver disease [36]. In liver pathology, TGF-β signaling is involved in every phase, spanning from initial injury to the development of inflammation, fibrosis, cirrhosis, and cancer [37]. While TGF-β aids in liver differentiation and regeneration, its excessive presence due to chronic damage triggers stellate cell activation, hepatocyte death, and ultimately, fibrosis and cirrhosis. Initially, during liver tumor formation, TGF-β may suppress growth, but prolonged signaling activation can later promote tumor progression [38,39].

In our study, liver toxicity was assessed using established gold-standard liver biomarkers, namely ALT and AST. The Food and Drug Administration (FDA) approved ALT use in 2009 for evaluating drug-induced liver toxicity (DILI), and it remains integral to the FDA-recommended approach for assessing DILI severity [40]. However, numerous novel liver-related biomarkers have emerged, such as cytokeratin-18, macrophage colony-stimulating factor receptor 1, and osteopontin [41,42,43]. The main disadvantage of these biomarkers, in contrast to miR-122, is that they are not liver-specific, and their fluctuations during ASCT may reflect the total impact of chemotherapy responses elicited on all tissues. Additional studies should be conducted to analyze their changes during conditioning chemotherapy and bone marrow transplant procedures, along with potential confounding factors affecting their levels.

Oxidative stress plays a significant role in drug-related liver injury by disrupting the balance between the production of reactive oxygen species (ROS) and the antioxidant defense systems to neutralize them [44]. This oxidative damage can result in various forms of liver injury, including hepatocyte apoptosis, inflammation, fibrosis, and ultimately, liver dysfunction. The administration of cytostatic agents, such as etoposide, melphalan, and carmustine, as part of conditioning regimens during bone marrow transplants, is a significant contributor to the disruption of redox balance and oxidative stress status in patients undergoing the procedure. This occurs through the increased production of free radicals and a reduction in host antioxidant defenses. Some reports indicated that myeloma and lymphoma patients undergoing ASCT displayed heightened oxidative activity, measured by malondialdehyde (MDA), a lipid peroxidation product, and catalase (CAT) and superoxide dismutase (SOD) enzymes along with a high DNA damage index, in comparison to the control group. Furthermore, the conditioning regimen worsened this state with subsequent return to baseline levels [45]. In another study, individuals undergoing bone marrow transplantation, whether autologous or allogeneic, exhibited indications of oxidative stress during the conditioning regimen. This was evidenced by reductions in plasma vitamin C levels and blood δ-aminolevulinate dehydratase (δ-ALA-D) activity, as well as an elevation in blood lipoperoxidation, observed in both groups of patients [46]. In another small study, urinary 8-hydroxydeoxyguanosine (8-OHdG) significantly increased immediately after conditioning therapy in allogenic and autologous bone marrow transplant recipients [47]. Moreover, high urinary 8-OHdG levels were associated with poor prognosis in patients undergoing allogenic bone marrow transplant. However, the oxidative stress imbalance is generally more visible in the latter phase of the bone marrow transplant, after the administration of conditioning regimen. In one study, there were no changes in pretransplant level and day O in oxidative stress markers including nonreversible protein oxidation indexed by protein carbonyls and 3-Nitrotyrosine and lipid peroxidation to 4-hydroxy-2-transnonal (HNE) [48]. Although an increase in oxidative stress has been reported in ASCT recipients, further assessment is necessary to determine its direct potential association with hepatotoxicity in this setting.

The main limitation of our study is evaluating only five preselected miRNAs based on the literature research. The more advanced experiments would involve high-throughput techniques involving miRNA sequencing or microarray analyses, enabling the identification of larger amounts of dysregulated miRNAs. However, this study primarily focused on validating miRNAs as biomarkers of liver injury in an ASCT clinical setting. The focus on serum miRNAs stems from the clinical availability, reliability, and ease of introducing blood-based biomarker tests to clinical practice. Serum miRNAs, released into circulation by various cell types, including hematopoietic and non-hematopoietic cells, would thus provide a more comprehensive insight into the incipient complications and outcome predictions [49,50,51]. In addition, assessment of miR-122 in other populations, including allogeneic hematopoietic stem cell transplant recipients, would be exciting in the context of hepatic GvHD, as miR-122 was already reported to be up-regulated in sera of patients with chronic GvHD [52]. The second main limitation of our study is limited sample size. We planned the sample size statistically to allow us the direct validation of our hypothesis and demonstration of the capabilities of miRNAs to be used in this setting. Secondly, it is important to highlight that patients undergoing ASCT are a very specific and small population, which makes recruitment difficult and time consuming. Expanding the sample size would necessitate a multicenter study collaboration. Despite the limited sample size, we believe our results are scientifically and clinically significant.

In conclusion, the high expression level of serum miR-122-5p as measured before the ASCT can potentially predict hepatotoxicity associated with the procedure. This observation could have practical implications for personalized medicine, potentially enabling more effective and cost-efficient management of patients undergoing ASCT with targeted monitoring or preemptive interventions.

## 4. Materials and Methods

### 4.1. Patients and Treatment

Patients who underwent autologous stem cell transplantation (ASCT) at the Department of Hematology and Transplantology, Provincial Multi-Specialized Oncology and Trauma Center in Lodz, Poland, were prospectively recruited for this study. Enrollment occurred between 2015 and 2022. The study adhered to the principles outlined in the Declaration of Helsinki and the principles of good clinical and laboratory practice. Informed consent was obtained from each patient for all procedures. Approval for all procedures was granted by the local ethical committee (The Ethical Committee of the Medical University of Lodz, No. RNN/424/19/KE). Inclusion criteria encompassed a minimum age of 18 years, a diagnosis of hematologic malignancy, and eligibility for ASCT as part of the treatment plan. Exclusion criteria included contraindications for ASCT, notably a Hematopoietic Cell Transplantation –Comorbidity Index (HCT-CI) exceeding 3. For patients with multiple myeloma (MM), the myeloablative conditioning regimen involved high-dose melphalan (200 mg/m^2^), while lymphoma patients received B(e)EAM. The latter regimen included carmustine (B) 300 mg/m^2^ or bendamustine (Be) 160–200 mg/m^2^ on day −6, etoposide 200 mg/m^2^ and cytarabine 200 mg/m^2^ twice daily from days −5 to −2, and melphalan 140 mg/m^2^ on day −1. From day +4 post-ASCT (72 h after transplantation) until engraftment, all patients received granulocyte-colony stimulating factor (G-CSF). Antimicrobial prophylaxis for each patient included ciprofloxacin 500 mg twice daily, acyclovir 800 mg twice daily, and fluconazole 400 mg once daily for those patients identified as high-risk for candidiasis.

### 4.2. Hepatotoxicity Assessment and miRNAs Selection

The liver toxicity was evaluated based on alanine transaminase (ALT) and aspartate transaminase (AST) laboratory tests performed during hospitalization. Levels of AST and ALT at the time of admission were set as a baseline for every patient. Each patient had ALT and AST evaluation during the study at least once in two weeks after ASCT. Using modified Seattle Criteria for Organ Toxicity after ASCT scale, we classified mild liver injury (grade 1) as an ALT > 2× baseline and ALT > ULN (Upper limit norm) or AST > 2× baseline and AST > ULN [53,54]. Similarly, moderate liver injury (grade 2) was classified as an ALT > 5× baseline and ALT > 3× ULN or AST > 5× baseline and AST > 3× ULN. Patients who met the criteria for liver injury in at least one evaluation within 14 days following hematopoietic stem cell transplantation (day 0) were assigned to the liver injury group. In this study, we selected for investigation miRNAs that were previously reported to be deregulated in liver injuries or associated with hepatocytes: miR-122-5p, miR-122-3p, miR-15b-5p, miR-99b-5p, miR-125a-5p [11,12,13,14,15,16,17].

### 4.3. Samples Collection and RT-qPCR

Blood samples were collected on the day of transplant (day 0), after the administration of the conditioning regimen and were drawn into anticoagulant-free containers and kept at room temperature for 30 min. The serum was separated by centrifugation at 1500× *g* for 15 min at 4 °C. The serum samples were kept at −80 °C until analysis.

Total RNA, including miRNA, was extracted from 200 µL volume of serum using the miRNeasy Serum/Plasma Advanced Kit (QIAGEN, Hilden, Germany), following a phenol-free protocol in accordance with the manufacturer’s instructions. Total RNA was eluted from the column using 20 µL of RNase-free water and stored at −20 °C until further use.

The expression levels of selected miRNAs were quantified by RT-qPCR using the miRCURY LNA miRNA Custom PCR Panels (QIAGEN). Briefly, 2 µL of complementary DNA (cDNA) was synthesized from the obtained total RNA including mature miRNAs (<200 bp), using the miRCURY LNA Reverse Transcription Kit (QIAGEN) according to the manual provided by the manufacturer. Two synthetic miRNAs, UniSp6 (0.075 fmol) and cel-miR-39-3p (0.001 fmol), were used as a positive control for cDNA synthesis. The cDNA was stored at −20 °C until further analysis.

Next, a premix of 3 µL of cDNA template (diluted 1:30), 5 µL of 2X miRCURY LNA SYBR Green PCR Master Mix, and filled with RNase-free water to the final volume of 10 µL, was aliquoted into the PCR plate. Real-time PCR was performed on a LightCycler480 II Real-Time PCR System (Roche, Pleasanton, CA, USA). The reaction was performed at 95 °C for 2 min, followed by 55 amplification cycles at 95 °C for 10 s and 56 °C for 1 min. Absolute quantification of miRNA was determined using the LightCycler^®^ 480 Software, Version 1.5 (Roche, Mannheim, Germany).

### 4.4. Statistical Analysis

The miRNA expression levels were determined using the ΔCt method. The RT-qPCR data were normalized by utilizing the mean expression value of two miRNAs within a particular sample, namely hsa-miR-27b-3p and hsa-miR-148b-3p. These miRNAs were identified as the most stable factors based on the analysis performed with NormiRazor Software (https://norm.btm.umed.pl/, accessed on 1 April 2023) in our prior study involving ASCT recipients [55,56]. The formula for calculating the normalized Ct values was: Normalized ΔCt = (mean Ct of hsa-miR-27b-3p and hsa-miR-148b-3p) − Ct miRNA of interest.

This method yields increased values for elevated miRNA expression, facilitating a clear interpretation of biomarker results.

Our research adhered to the predetermined sample size calculation. Anticipating a difference of 0.8 ΔCt between the liver injury group and control means, with a standard deviation of 1 ΔCt and an experimental to control ratio of 1:1, we determined that examining 26 subjects in each group was necessary to potentially reject the null hypothesis of equal means between experimental and control groups with a power of 0.8 and a type I error probability of 0.05.

Unsupervised hierarchical cluster analysis was performed using Morpheus software (https://software.broadinstitute.org/morpheus, accessed on 12 December 2023) with the complete linkage method and 1-minus Spearman correlation metric. ROC AUC analyses were performed to evaluate the diagnostic ability of liver injury of analyzed miRNAs. Univariate and multivariate logistic regression analyses were conducted to establish the association of examined miRNA and clinical variables with liver injury occurrence after ASCT during the early post-transplant period—since day 0 up to 14 days after transplant. All statistical analyses were performed using Statistica Version 13.1 (TIBCO, Palo Alto, CA, USA). *P* Values lower than 0.05 were considered statistically significant.

The target genes of miRNAs included into the multivariate model construction were predicted using miRNet 2.0 (http://www.mirnet.ca/, accessed on 1 April 2024) [57]. Using the miRNet and miRTarBase v8.0, an enrichment analysis of KEGG pathways was conducted to investigate the role of selected miRNAs in biological processes and signaling pathways.

## Figures and Tables

**Figure 1 ijms-25-04355-f001:**
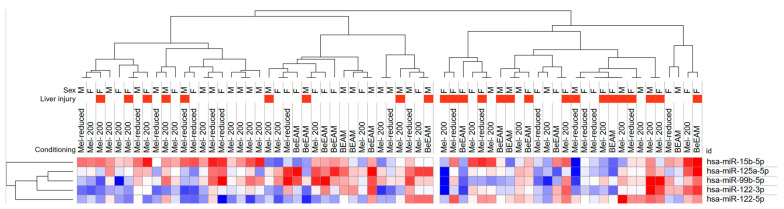
Heatmap with unsupervised hierarchical clustering analysis of serum samples autologous hematopoietic stem cell transplant (ASCT) recipients according to microRNA (miRNA) expression levels. The dendrogram shows two clusters of patients according to the miRNA expression profile. The cluster with higher miR-122-5p and lower hsa-miR-125a-5p expression tends to experience liver toxicity more frequently. One minus Spearman correlation distance metric and average linkage method were used in hierarchical clustering.

**Figure 2 ijms-25-04355-f002:**
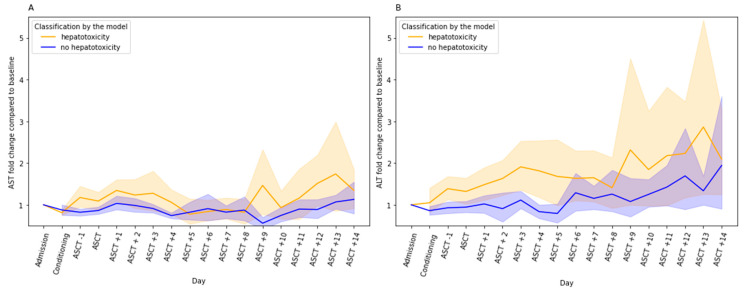
Aspartate transaminase (AST) (**A**) and alanine transaminase (ALT) (**B**) fold changes (FC) during autologous hematopoietic stem cell transplantation in relation to the class assigned by the 10-fold cross-validated model utilizing two miRNAs serum levels—miR-122-5p and miR-125a-5p—on day 0 (before transplantation). The lines indicate FC with a 95% confidence interval compared with baseline (at admission) AST/ALT levels.

**Figure 3 ijms-25-04355-f003:**
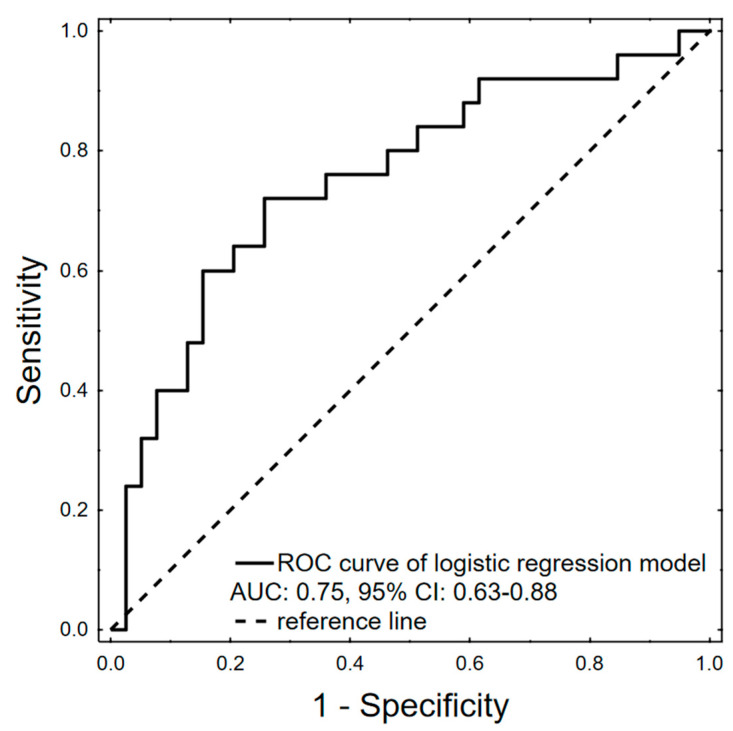
ROC curve of logistic regression model after 10-fold cross-validation predicting hepatotoxicity occurrence during autologous stem cell transplantation, AUC: 0.75, 95% CI: 0.63–0.88, *p* = 0.0001.

**Table 1 ijms-25-04355-t001:** The characteristics of patients included in the analysis. Data are shown as frequencies—N (%)—unless otherwise specified.

Characteristics	Total N (%)
Number of patients	66 (100)
Sex	F: 34 (51.5)
	M: 3 2 (48.5)
Age at ASCT	59
Median (IQR)	(49–65)
Diagnosis	
Multiple Myeloma	50 (75.8)
Hodgkin Lymphoma	8 (12.1)
Mantle Cell Lymphoma	4 (6.1)
Diffuse Large B-cell Lymphoma	2 (3.0)
Angioimmunoblastic T-cell Lymphoma	1 (1.5)
Anaplastic Large Cell Lymphoma	1 (1.5)
Myeloma stage	
ISS I	17 (34.0)
ISS II	10 (20.0)
ISS III	15 (30.0)
Missing	8 (16.0)
Ann Arbor	
I	1 (8.3)
II	2 (16.7)
III	4 (33.3)
IV	5 (41.7)
Conditioning	
Mel-200	32 (48.5)
Mel-reduced	18 (27.3)
BeEAM	11 (16.7)
BEAM	5 (7.6)
HBV/HCV infection	
Without HBV/HCV	58 (87.9)
HBV	7 (10.6)
HCV	1 (1.5)
Markers of liver injury preASCT	
Median (IQR)	
AST (U/L)	19 (16–22)
ALT (U/L)	20 (14–28)
Bilirubin (mg/dL)	0.36 (0.3–0.5)
Albumins (mg/dL)	41.25 (38.3–44.4)
GGTP (U/L)	18 (13–33)
ALP (U/L)	79.5 (59.5–105.5)
Antifungal treatment after ASCT	
No	34 (51.5)
Yes	30 (45.5)
Missing	2 (3)
Abnormal ALT and/or AST at admission	
No	64 (97.0)
Yes	2 (3.0)

Abbreviations: ALP—Alkaline Phosphatase; ALT—Alanine Aminotransferase; ASCT—Autologous Stem Cell Transplantation; AST—Aspartate Aminotransferase; BEAM—Carmustine, Etoposide, Cytarabine, Melphalan; BeEAM—Bendamustine, Etoposide, Cytarabine, Melphalan; GGTP—Gamma-Glutamyl Transpeptidase; HBV—Hepatitis B Virus; HCV—Hepatitis C Virus;IQR—Interquartile Range; ISS—International Staging System; Mel-200—High-dose Melphalan.

**Table 2 ijms-25-04355-t002:** Univariate logistic regression for hepatic toxicity (HT) during hospitalization after ASCT. miRNA expression levels as continuous variables after the conditioning chemotherapy were included in the analyses.

Variable	Value	OR	95% CI Lower	95% CI Upper	*p*-Value
HBV/HCV	No	*Reference*			
	Yes	0.20	0.02	1.75	0.1471
Diagnosis	Lymphoma	*Reference*			
	Myeloma	0.52	0.16	1.61	0.2547
Sex	Male	*Reference*			
	Female	1.34	0.49	3.63	0.5696
Age at ASCT	<60 years	*Reference*			
	≥60 years	0.37	0.13	1.08	0.0679
Albumin	g/L	1.09	0.92	1.29	0.3244
AST	IU/L	1.00	0.93	1.07	0.9978
ALT	IU/L	1.01	0.97	1.05	0.7740
Bilirubin	Above median	0.44	0.14	1.39	0.1636
ALP	IU/L	1.00	0.98	1.02	0.7331
GGTP	IU/L	1.02	0.98	1.07	0.2316
Stage	ISS ≤ 2 or Ann Arbor ≤ III	*Reference*			
	ISS 3 or Ann Arbor IV	0.61	0.20	1.87	0.3864
Antifungal treatment	No	*Reference*			
	Yes	0.71	0.26	1.95	0.5112
Previous lines of treatment	<2	*Reference*			
	≥2	1.44	0.53	3.94	0.4749
Conditioning regimen	Mel-200	*Reference*			
	Mel-reduced	0.66	0.19	2.35	0.5207
	B(e)EAM	1.58	0.47	5.35	0.4595
hsa-miR-15b-5p	1 dCt	0.49	0.18	1.36	0.1717
hsa-miR-125a-5p	1 dCt	0.56	0.30	1.05	0.0686
hsa-miR-99b-5p	1 dCt	0.69	0.37	1.28	0.2442
hsa-miR-122-5p	1 dCt	1.59	1.09	2.32	0.0164
hsa-miR-122-3p	1 dCt	0.86	0.60	1.23	0.3990

1 dCt—a difference of 1 amplification cycle (Ct) between the average Ct of the two reference miRNAs and the miRNA of interest. Abbreviations: ALP—Alkaline Phosphatase; ALT—Alanine Aminotransferase; ASCT—Autologous Hematopoietic Stem Cell Transplantation; AST—Aspartate Aminotransferase; B(e)EAM—carmustine (B) or bendamustine (Be), etoposide, cytarabine, and melphalan; CI—Confidence Interval; GGTP—Gamma-Glutamyl Transferase; HBV—Hepatitis B Virus; HCV—Hepatitis C Virus; Mel—melphalan, OR—Odds Ratio.

**Table 3 ijms-25-04355-t003:** Multivariate logistic regression analysis for hepatic toxicity (HT) during hospitalization after ASCT.

Variable	Value	OR	95% CI Lower	95% CI Upper	*p*-Value
Sex	Male	*Reference*			
	Female	1.91	0.56	6.50	0.3029
Age at ASCT	<60 years	*Reference*			
	≥60 years	0.29	0.07	1.24	0.0941
hsa-miR-122-5p	1 dCt	2.10	1.29	3.42	0.0029
hsa-miR-125a-5p	1 dCt	0.27	0.11	0.71	0.0079
Conditioning regimen	Mel-200	*Reference*			
	Mel-reduced	0.64	0.13	3.15	0.5832
	B(e)EAM	1.79	0.37	8.63	0.4699

1 dCt—a difference of 1 amplification cycle (Ct) between the average Ct of the two reference miRNAs and the miRNA of interest.

## Data Availability

The data generated during the current study are included in the article/Appendix A.

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
