# Peer review of "Serum Levels of miR-122-5p and miR-125a-5p Predict Hepatotoxicity Occurrence in Patients Undergoing Autologous Hematopoietic Stem Cell Transplantation"

_ijms, 2024, doi:10.3390/ijms25084355_

Round 1

Reviewer 1 Report

Comments and Suggestions for Authors

This is a well-designed and performed study. It is an original study. It well recognized in the literature that hepatotoxicity is a leading complication in recipients of autologous hematopoietic stem cell transplantation (ASCT). Therefore, the search for biomarkers of liver damage is clearly justified. The main question of the present research was to evaluate the potential role in the prediction of liver injury of five selected miRNAs (miR-122-5p, miR-122-3p, miR-15b-5p, miR-99b-5p, miR-125a-5p) in the setting of ASCT. This study is relevant because liver damage is a factor of mortality and morbidity in patients undergoing hematopoietic stem cell transplantation. Presently, there are no studies evaluating the role of these miRNas in the liver damage in the setting of ASCT. Therefore, this study is a relevant contribution to the field because liver damage is a factor of mortality and morbidity in these patients. The relevance of the present study is also evident based on the fact that the markers commonly used to measure hepatic damage (albumin, ALP, bilirubin, AST, GGTP, and ALT) were not associated with ASCT-related hepatotoxicity.

The results are well described in a meticulous way in three tables and three figures. The characteristics of the studied patients, including their underlying diseases, are presented in detail.

The methods used to evaluate hepatotoxicity, miRNA selection, sample collection and RT-qPCR are described in sufficient detail to be reproduced in another lab.  In particular, the methods used to analyze results are described in detail in the “statistical analysis” section.

The authors state the study limitations related to the fact that preselected miRNA was based in the literature search

The conclusion is supported by the data presented. They found that serum levels of miR-122-5p and miR-125a-5p, at the day of transplant, were identified as independent prognostic factors of hepatotoxicity occurrence.

Inglés

….as an independent prognostic factors….should be …as independent prognostic factors…

Major

Lines 69-70: The following sentence “…. all previously reported to be deregulated in the drug-induced liver injury, in the specific setting of ASCT.” should be supported with references

Liver hepatotoxicity was evaluated exclusively using the values of ALT and AST obtained sequentially during hospitalization. Please discuss if other potential markers (not the traditional markers) could be useful under the studied conditions.

The authors should consider include in the study limitations the low number of the patients studied.

Inflammation and oxidative stress markers were not measured. Please discuss the potential role of these variables in the hepatotoxicity developed by these patients.

Please discuss

Minor

It is hard to see Figure 1.

Figure 2. It is hard to see the letters used inside it and in both axis.

I was unable to find Figure S1.

Abbreviations should be used consistently and correctly. Examples

Line 20. Define miRNAs.

Line 25. Define RT-QPCR.

Line 27. Define OR y CI.

Line 30. Define ROC.

Line 60. Define RNA.

Line 78. Define IgG.

Line 81. Define IV.

Line 89. Define HBV y HCV.

Line 91. Define ALT y AST.

Tabla 1. Define IQR in the Table footnote.

Figure 1:

          Line 112. Define ASCT en el pie de figura.

          Line 113. Define miRNA.

Table 2. All the abreviations should be defined in the table footnote.

Figure 3:

          Define 154. Define ROC.

          Define 155. Define AUC, and CI.

Line 267. Define cDNA.

Line 87 BeEAM is defined as “Bendamustine, Etoposide, Cytarabine, Melphalan”. Plase homogenize because in lines 130, and 235 is used as B(e)EAM. In addition, In the line 188 is used as  B(e)/EAM.

Comments on the Quality of English Language

….as an independent prognostic factors….should be …as independent prognostic factors…

Reviewer 2 Report

Comments and Suggestions for Authors

This is a well-written paper that suggests that serum levels of specific miRs may predict liver toxicity after autologous hematopoietic stem cell transplantation. However, because the number of analyzed cases is small and the diseases and patient backgrounds involved are diverse, the study should be carefully studied to draw firm conclusions.

1. Fig. 2 seems to show the relationship between the amount of hsa-miR-122-5p and liver damage. In that case hsa-miR-122-5p should be mentioned in the figure legend.

2. Statistical analysis methods should be well explained in the method section.

3. Has it been confirmed that there is no confounding among the factors selected for multivariate analysis?

4. Myeloma patients accounted for 76% of the patients. The authors should analyze whether similar results can be obtained by analyzing only myeloma patients.

5. It should be briefly described how miR measurement can help prevent and manage post-autotransplant liver damage.

6. Were the miRs analyzed this time not related to the increase in bilibin after transplantation?

Author Response

Please see the attachment,

Reviewer 3 Report

Comments and Suggestions for Authors

The manuscript by Mikulski, et al entitled “Serum levels of miR-122-5p and miR-125a-5p predict hepatotoxicity occurrence in patients undergoing autologous hematopoietic stem cell transplant” examines the role of five mi-RNA’s on the development of liver toxicity after autologous stem cell transplant.  Each of the mi-RNA examined have been identified to be involved in liver toxicity in other clinical conditions.  Hepatotoxicty occurs in up to 40% of patients after autologous stem cell transplant.  The authors show that both mi-122-5p and miR-125a-5p regulate hepatoxicity in opposite directions in these patients.  Multivariate logistic regression models were used to confirm these results. 

Overall, the results appear to be valid.  Concerns regarding the manuscript is that it is descriptive in character and lacks novelty as these mi-RNA’s have been shown to involved in hepatotoxicity in other clinical conditions.  Potential mechanisms for how miR-122-5p and miR-1215a-5p regulate hepatotoxicity after autologous stem cell transplant are not fully explored with only a brief discussion provided and no extensive literature review.

Round 2

Reviewer 2 Report

Comments and Suggestions for Authors

The authors have well responded to the comments. 

Reviewer 3 Report

Comments and Suggestions for Authors

All concerns from the original review were addressed in the revision